# Public Perception of the First Major SARS-Cov-2 Outbreak in the Suceava County, Romania

**DOI:** 10.3390/ijerph18041406

**Published:** 2021-02-03

**Authors:** Daniel Lucheș, Despina Saghin, Maria-Magdalena Lupchian

**Affiliations:** 1Department of Sociology, West University of Timișoara, 300223 Timisoara, Romania; daniel.luches@e-uvt.ro; 2Department of Geography, Stefan cel Mare University of Suceava, 720229 Suceava, Romania; mmlupchian@gmail.com

**Keywords:** COVID-19, health crisis, Romania, public perception

## Abstract

The first months of 2020 were marked by the rapid spread of the acute respiratory disease, which swiftly reached the proportions of a pandemic. The city and county of Suceava, Romania, faced an unprecedented crisis in March and April 2020, triggered not only by the highest number of infections nationwide but also by the highest number of infected health professionals (47.1% of the infected medical staff nationwide, in April 2020). Why did Suceava reach the peak number of COVID-19 cases in Romania? What were the vulnerability factors that led to the outbreak, the closure of the city of Suceava and neighboring localities, and the impossibility of managing the crisis with local resources? What is the relationship between the population’s lack of confidence in the authorities’ ability to solve the crisis, and their attitude towards the imposed measures? The present article aims to provide answers to the above questions by examining the attitudes of the public towards the causes that have led to the outbreak of an epidemiological crisis, systemic health problems, and the capacity of decision makers to intervene both at local and national level. The research is based on an online survey, conducted between April and May 2020, resulting in a sample of 1231 people from Suceava County. The results highlight that the development of the largest COVID-19 outbreak in Romania is, without a doubt, the result of a combination of factors, related to the medical field, decision makers, and the particularities of the population’s behavior.

## 1. Introduction

The SARS-CoV-2 pandemic has seen a sudden and troubling global spread [1,2,3]. Since December 2019, when the first cases were registered in the Wuhan region of China, the number of reported infections has increased significantly, exceeding 80 million at the end of December 2020 [4].

The responses of authorities and health systems in various states, aiming to decelerate the spread of the virus and decrease the number of infections, have been more or less effective. The speed with which the first measures were implemented, the level of awareness towards the danger, and the protection capacity of the medical staff proved to be decisive factors in halting the spread of the pandemic. Furthermore, the overall quality of the health infrastructure, the ability of states to impose and reinforce restrictive measures, as well as the responsibility and behavior of the population that manifests itself differently at national and regional levels, are all determinants that play a crucial role in the efforts to manage the rapid growth of infection rates. Given the global efforts that have been directed towards understanding and managing this pandemic, the scientific literature in the field is rather abundant, with numerous studies addressing the link between various categories of factors—demographic, socio-economic, cultural, environmental—and the impact of the COVID-19 pandemic on the population [1,5,6,7,8,9,10,11]. One of the main conclusions that can be drawn from existing research is formulated by Cruz et al. [9], who argue that “there is no easy answer to why one area should experience COVID-19 differently from another”. Studies addressing the impact of various factors on the spread of pandemic and its effects on society are preceded by similar analyses, conducted in the context of other epidemics that have affected humankind (avian influenza, swine flu, SARS, MERS). These analyses [12,13,14] can be used to construct a model for the current events, and provide an opportunity to examine the role that the experience of a pandemic has on the efficacy of a society’s reaction in a similar situation [9]. Anxiety generated by the risk of illness, its impact on the cognitive and emotional state of individuals, and the mental health challenges that arise as a result are also addressed extensively in the existing literature [15,16,17,18]. Last but not least, efforts geared towards identifying the weaknesses of economic and social systems that have inhibited the ability to respond efficiently to the pandemic have also generated valuable insights [10,19,20].

In the specific context of Romania, the proportions of the epidemic have aggravated existing problems of a health system that is characterized by systemic underfunding, corruption, and a strong exodus of white robes [21]. The first case was confirmed on 26 February 2020, in Gorj County, and the number of reported cases rapidly increased to 1000 within a month, escalating to over 10,000 during the first two months, and reaching 300,000 confirmed cases at the beginning of November 2020, according to data provided by the Strategic Communication Group within the Ministry of Internal Affairs (GCS) [22].

The development of the coronavirus epidemic in Romania was marked by the emergence of a major outbreak in the municipality and county of Suceava, in north-eastern Romania. The situation posed several challenges, not only due to the exceptionally large number of cases registered amongst the population, but especially due to the extremely high number of illnesses amongst medical staff. In the context of the measures adopted by European member states and in order to prevent extensive community transmission, a state of emergency was declared on the 16th of March in Romania. Despite all the restrictive measures imposed at the national level, an extremely grave situation continued to develop in Suceava County where, in mid-April, 24.8% of all national reported cases were observed (GCS) (Figure 1) The functional capacity of the County Emergency Hospital (SCH) in Suceava was reduced, whilst the sectors in which COVID-19 patients were admitted, the Emergency Reception Unit, the laboratories, and the administrative and logistical sectors operated at a “breakdown” capacity due to low staff numbers, functioning with approximately 15% of the hospital staff coverage.

In attempts to efficiently manage the epidemiological disaster that was developing in Suceava, the city and eight neighboring areas were quarantined through a military ordinance on the 31st of March 2020; a measure which affected approximately 200,000 people (Figure 1). The local population suffered a major psychological shock as a result, generating panic and uncertainty. For the first time in 30 years, the population’s freedom of movement was restricted, and their fundamental rights constrained, rights Romanians had regained as a result of the 1989 Revolution. The symbolic meaning of these measures was powerful, particularly in a country with a fragile democratic regime. Furthermore, the psychological effect of these measures was amplified by their overlap with Easter holidays, in a region where religious practices, paired with a strong attachment to Christian values and traditions, are still strongly adhered to. Between the 2nd and 30th of April, due to the difficulty in managing this epidemiological outbreak, the planning deficiencies in the reaction and protection against the virus, and the critical absence of specialised human resource, the leadership of the county hospital was replaced by a military team.

Despite the restrictions imposed during the lockdown period, other small hospitals continued to be closed in the county. In the following two weeks, the number of reported cases increased at an alarming rate, with Euronews journalists classifying Suceava County as the most critical epidemiological hotspot in Romania, and national sources nicknaming the region “Romania’s Lombardy”. On April 16th, Suceava County had 24.8% of the total cases of Covid-19 in Romania, and 47.1%. of the total number of infected healthcare professionals reported nationally. Regardless of the measures implemented, the county continued to report the highest number of cases per 1000 inhabitants, even after the country’s capital was ranked first in total number of diseases, in mid-July.

The present research aims to provide answers to a series of questions revolving around the epidemiological crisis that developed in March and April 2020, in Suceava County. Beginning from the premise of the crisis’ gravity, which will also be the subject of a criminal investigation, the following questions will be addressed: Why did the city and county of Suceava reach the highest number of national infections in Romania, only a few weeks after the first reported case? Why did the burst of infections (both amongst the population and medical staff) generate an unprecedented health crisis, despite the city benefitting from an ultramodern hospital, recently renovated using European funds? How did the population of the county perceive the situation? How did they relate to the drastic restrictions imposed by the authorities? What is the relationship between the population’s lack of confidence in the authorities’ ability to manage and resolve the crisis, and their attitude towards the measures imposed?

The attitudes of the population were analyzed according to various sociodemographic variables (age, residence—belonging or not to the quarantined area, gender, etc.), factors which allow the construction of a clearer overall understanding of the crisis. The most frequently used tool for analyzing population perception and behavior is the KAP (Knowledge, Attitude, Practice) model, through which information about what a portion of the population knows, thinks, and does in a given context can be collected. Analyses of this type have been extensively used both during other epidemics—Influenza, SARS, MERS, Zika, etc. [5,23,24] and in the context of the current pandemic [11,17,25,26,27,28,29,30,31]. The relationship between various sociodemographic characteristics (age, gender, level of education, occupation) has been analyzed both in the context of past epidemics [12,32,33,34,35] as well as in the current epidemiological context [11,36,37,38]. There are few studies that focus specifically on the spread and impact of the pandemic in Romania, focusing either on the perception and attitudes of the population towards the threat of the disease and the restrictions imposed by the authorities [39], or on changes in consumer behavior [40,41,42]; the economic impact [43,44] as well as the impact on the educational system have also been previously addressed [45].

Within this context, the present study will frame a foundational approach for future research addressing the ways in which authorities reacted to the issues arisen as a result of the crisis in Suceava. Given the extent of the crisis at a global level, the public opinion towards the imposed restrictive measures, and the behavioral changes captured in the present research, observed at the beginning of the crisis in Suceava, can be further compared with perceptions, attitudes and manifested behaviors that can arise as a result of such measures over a longer period of time.

## 2. Materials and Methods

The present paper uses data collected in the first two weeks of May 2020 through an online survey, which targeted the population living in Suceava County. The aim of the research is to uncover the perception of the population towards the causes of the COVID-19 pandemic in Romania, the actions of local and national authorities, the behavioral changes generated by local decisions gearing society towards a new reality, and the attitudes towards the restrictions and decisions assumed by authorities.

The survey link, hosted on Google Forms, was promoted through recommendation from students, teachers, and friends. In order to increase the participation rate and ensure a wider distribution, the snowball method was employed, asking each respondent to distribute the questionnaire to acquaintances. The questionnaire was comprised of 33 questions grouped in distinct research topics to address issues such as the perception of the causes underlying the development of the pandemic in the region, the public attitude towards the responses of authorities and local decision makers, the behavioural changes that occurred during the period of interest, the acceptance and/or rejection of restrictive measures imposed by the emergency ordinance, the perception of the impact of the new reality at individual and community level, and the public attitude towards the way in which civil society has been involved in reducing the impact generated by the crisis.

Initially, 1273 questionnaires were completed, but after the validation stage, 1231 questionnaires representing the volume of the studied sample were selected. The questionnaires in which the age of participants was below 18 years old and the questionnaires that had respondents from other counties were eliminated.

Structurally the sample included 67.3% women and 32.7% men; 75.5% of the respondents reported residing in the urban areas of the county and 24.5% in rural areas, of which 65.7% live in Suceava (county capital) and 34.3% live in other towns or villages from the Suceava county; the mean age of the sample is 38.05 years, highlighting a strong representation of the 25–50 age group (64.9% from the sample). Regarding the status of the population after involvement in economic activities, 77% of respondents are included in active population (employed in private or state companies, entrepreneurs, practitioners of liberal professions), respectively 23% inactive population (pupils, students, pensioners, households).

All variables used in our research are detailed as follows:*Age of respondents*: this was measured with a single question asking respondents to indicate their age. Subsequently, three age categories were created: <25 years (coded 1), 25–50 years (coded 2) and >50 years (coded 3);*Causes*: dichotomous variable (1—internal causes and 2—external causes), obtained by recoding the following response variants: *internal causes* included answers: inadequate management of the Suceava County Hospital (SCH), the lack of responsibility of medical staff and lack of protective equipment for hospital staff. Moreover, *external causes* included inappropriate behavior exhibited by Romanian patients who returned from abroad;*Quarantined locality*: refers to the situation of the locality, if was included in the quarantine or not; dichotomous variable (1—yes and 2—no);*Authorities*: refers to the authorities that have adopted various measures to reduce the effects of SARS-Cov-2 outbreak (1—local authorities, 2—national authorities);*Trust*: refers to confidence in the medical staff after the removal of the military leadership; dichotomous variable (0—distrust and 1—trust);*Factors*: refers to the factors that contributed decisively to the onset of the current crisis in Suceava County. For each factor there were dichotomous answers (1—to a very small extent and to a small extent, 2—to a large extent + to a very large extent). In the results section appear only the factors for which there were statistically significant associations with the status of the locality (quarantined locality or not);*Local reactions*: refers to the way in which the participants of the study perceived the reaction of the local authorities to the challenges of the pandemic; the following codes were used: 1—They were overwhelmed by the situation, and did not react accordingly, 2—They reacted normally, given the crisis conditions, 3—They reacted promptly, trying to intervene to the best of their ability.*National reaction;* refers to the way in which the participants of the study perceived the reaction of the national authorities to the challenges of the pandemic; codes similar to those from local reactions were used;*Restrictions*: restrictions: refers to the level of difficulty of complying with the restrictions imposed by the authorities. For each of the restrictions a Likert scale was used, where 1—very difficult, respectively 4—not at all difficult.

## 3. Results

### 3.1. Factors that Determined the Outbreak in Suceava

The development of the outbreak in Suceava cannot be linked to one singular determinant but is the result of a cumulation of factors, connected with both the medical and the political field. Furthermore, the particularities, behavior and attitudes of the local population have also been observed to play a crucial part in the effective management of the crisis. The perception of the local population was analyzed using a variety of sociodemographic variables, including but not limited to age, gender, and residence, factors which allow for the establishment of an overview of the epidemiological crisis.

Understanding the factors that contributed to the emergence and development of the first major outbreak of COVID-19 in Romania requires a differentiated approach/analysis. Within this context, the results of the present paper outline two categories of determinants: external ones, associated with the inappropriate behavior exhibited by Romanian patients who returned from abroad, particularly from areas with a high incidence of reported cases, and internal causes that have indirectly contributed to the outbreak, such as the inadequate management of the Suceava County Hospital (SCH), the role of the Suceava County Council as a tutelary authority, the weak involvement of the Public Health Directorate and the lack of responsibility of medical staff. In the specific context of perceptions shared by the present sample, one of the most important identified factors was external, represented by the return of a large number of Romanians from countries in which the incidence of illness was high, paired with the non-declaration of their arrival from the affected areas (38.1%). Complementary to the source that generated the first incidences of infection, respondents acknowledged that the development of the outbreak could also be linked to internal causes (61.9%), represented by the poor management of SCH. The inadequate behavior of medical staff and the lack of protective equipment was identified as a secondary cause, with much lower attitude shares of 15.8% and 11.2%, respectively.

If the return of Romanian migrants is disregarded as the main factor that contributed to the spread of the pandemic in the region, it can be observed that more than a third of the respondents believe that the inefficient management of the outbreak, paired with inadequate local decisions, played a decisive role in the exponential increase of reported cases. At a closer inspection, no hospital management decision, epidemiological recommendation, or ongoing practice could be identified, that ensured the sizing and separation of specific functional circuits. Furthermore, there was an apparent lack of risk management approaches implemented outside the hospital, and the testing capacity was reduced, failing to cover the investigative needs within a reasonable timeframe. Effective communication was problematic, the test results were delivered late (particularly in relation to the needs), and the supply of medical services was inadequate in the context of specific demands posed by the COVID-19 crisis. This reality is reflected in the insufficient number of health workers, and in the difficulty of providing prompt medical services of higher standards. Moreover, the investigation into the outbreak in Suceava revealed that the protection of personnel was insufficient, inadequate, and mostly improvised.

The public perception addressed in this study highlights three distinct factors that have contributed to the emergence of the crisis in Suceava county (Kaiser-Meyer-Olkin Measure of Sampling Adequacy = 0.729, *p* < 0.00) as can be observed in Table 1:Factor 1—the political management of the county and the leadership of the county hospital accounts for 25.1% of the observed varianceFactor 2—the inadequate behavior of citizens and the lack of sanctions explain 23,6% of the varianceFactor 3—the unprofessional attitudes of medical staff explain 23.2% of the variance.

In addition to the opinions based on which the above contributing factors were identified, the participants also expressed concern towards the increasing number of medical staff who were contaminated in SCH, especially given that at the time of the study, approximately half of the total number of infected medical professionals in Romania were activating in this institution. The analysis of the answers provided by the respondents contributed to the structuring of two major factors that could explain the high incidence of infection amongst healthcare professionals (Kaiser-Meyer-Olkin Measure of Sampling Adequacy=0.773, *p* < 0.00, Table 2):Factor 1—the lack of professionalism exhibited by medical staff explains 36.9% of the varianceFactor 2—the consternation of medical staff as a result of dealing with an unprecedented contemporary crisis accounts for 24.5% of the variance

One of possible explanations related to the development of the outbreak in Suceava can be determined by the age of the respondents. In this sense, we noticed the existence of statistically significant differences between the age of respondents t(1229) = 2.817, *p* = 0.005 who appreciate that the outbreak has spread due to internal causes (M = 38.8 years, SD = 12.1; improper management of the hospital, non-compliance with protection measures of medical staff or even the lack of protective equipment) compared to those who appreciate that the uncontrolled development of the crisis is due to external causes (M = 36.9 years, SD = 11.6; return of people from areas affected by the pandemic). In this way, we can observe that people with a higher average age appreciate that the situation is due more to the elements related to the local factor (management, medical staff and available resources) compared to the return of migrants from areas with high incidence of infections with Covid-19.

The importance associated with the contributing factors fluctuates depending on the respondents’ residence and their proximity to or distance from the outbreak. Therefore, the inhabitants of the quarantined area, located in close proximity to the municipality of Suceava, display a strong tendency to blame the medical staff, whilst those residing outside the quarantine area allocate more importance to general issues related to systemic deficiencies visible in the Romanian medical system, bribery, the underfunding of the system or even the lack of adherence of citizens to restrictive measures (Table 3).

### 3.2. Trust towards Authorities and the Positioning of the Population in Relation to the Implemented Measures

Understanding the way in which the public perceives a threat and positions itself in relation to the trust exhibited towards authorities and the measures imposed by them can contribute to the effective management of crises as a result of a pandemic. The experience gathered throughout this year highlights the need and importance of hospital leadership management training, and the establishment of a trustworthy logistical system for emergency situations [46].

The perception of the public towards the reaction of local and central authorities following the emergence of the outbreak in Suceava county highlights a more lenient opinion trend towards central authorities, in comparison to local ones. Such a trend consolidates the indirect responsibility and accountability of local authorities in managing the crisis through decisions and actions implemented throughout the period of interest. It can be observed that over 73% of the participants believe that local authorities were overwhelmed by the crisis, a finding which could be interpreted as a lack of trust towards the implemented measures and the reaction of decision-makers, χ^2^(2) = 469.078, *p* = 0.00. Similarly, it can be observed that trust in the central authorities highlights a higher level of trust as a result of radical decisions implemented to manage the outbreak of COVID-19. The appointment of a military leadership to the SCH was believed to be adequate by over 88% of participants, framing the army as an institution that generates feelings of respect and safety amongst the public. Moreover, the lockdown of the city of Suceava and the neighboring localities was considered appropriate by 78% of the study participants.

In the context of the decision to direct non-COVID patients to other hospitals in the county and in the region, it can be observed that, despite the measure being positively valued by 69% of respondents, it generated a limited support when compared to other decisions of the central administration. This finding determines that the implementation of such a decision results in creating a climate of apprehension and fear amongst people with multiple health concerns, who feel deprived of access to quality health care, and the disagreement towards this measure was primarily shared by the active population, aged between 25 and 50, χ^2^(2) = 6.042, *p* = 0.04. Beyond the local particularities of the context, the results also highlight the existence of a link between the perception towards the reactions of the local and central authorities, a finding that outlines the emergence of two diametrically opposed attitudes. The presence of an unfavorable attitude can be observed both towards local and national authorities, a radical opinion that is shared by more than one third of the total respondents (37.5%), emphasizing a disapproving attitude towards any type of intervention that the authorities at both levels would have made. In comparison, a positive opinion was expressed towards the local or central intervention of authorities by over 21% of the participants, which could be explained by the public’s concern with regards to the severity of the situation.

The residence of the participants was observed to form an important consideration that contributed to shaping the negative attitude of the sample towards the local authorities. The findings highlight that, when respondents live in one of the locked down localities, a tendency towards negative perceptions of the local decision-makers emerges despite the fact that the decision to quarantine the area was ultimately imposed by central authorities, rather than local ones χ^2^(2) = 32.936, *p* = 0.00.

In the context of the measures implemented by authorities to limit the spread of the virus, the present paper assessed the level of difficulty perceived by the participants using a Likert-type scale, (1—very difficult, 4—not at all difficult). The results show that respondents perceived the restrictions related to spatial mobility and the limitation of possibilities to meet with family or friends more difficult (Table 4). On the opposite spectrum, measures regarding the use of protective equipment (gloves, masks) in public spaces and the restricted access to shopping (due to the closure of commercial spaces and galleries) have been accepted with less resistance by the public.

As expected, the public attitude towards the measures adopted by authorities was found to be directly influenced by the reported residence of the participants (quarantined locality or not). The measures aimed at limiting recreational activities, sports and travel outside the quarantined area were perceived to be more difficult by participants residing in quarantined localities, when compared to the perceptions of respondents residing in areas where spatial mobility was not restricted (Table 5). Furthermore, for respondents residing in areas where the freedom of movement was not restricted, the primary concern stemmed from restrictive measures prohibiting groups bigger than 3–4 to meet and gather or the limited access to “non-essential” shopping (considering that Suceava, the county seat, is the main commercial hub in the area).

Compliance with the restrictive measures imposed by authorities was perceived differently by the participants. Young participants (up to 25 years old) admitted to finding it difficult to limit their shopping trips as much as possible, particularly in a context in which this activity can be a way to spend time with people in physical and social proximity. Moreover, the limited travel for fitness activities was perceived as more difficult by active people, aged between 25 and 50 years, whilst participants over 50 found the wearing of masks problematic. This particular concern can stem from the presence of a medical context and the likelihood of associating the wearing of a protective mask with potential breathing difficulties during activities carried out in public spaces (Table 6).

Despite the massive impact that the health crisis in Suceava County had on the local community, it can be observed that the trust levels of the public towards the hospital’s medical staff reached comparable values between the two analysed moments—before the crisis and after the departure of the military management of the hospital, χ^2^ (1) = 349.25, *p* = 0.00.

### 3.3. Adjusting the Public’s Behaviour

The behavior of the population during the situation that developed as a result of the pandemic is both an indicator of the existing level of concern and a measure of the degree of confidence in the decisions taken by the authorities in an attempt to manage an unprecedented crisis. 95% of the participants stated that they completely or almost completely comply with the restrictions imposed during the period of interest. Despite the share of those who comply with the restrictive measures being particularly high, a variety of factors that allow for a differentiation between the public’s level of compliance have been identified. Women (M = 4.6, SD = 0.6) show a tendency to be more compliant when compared to men (M = 4.4, SD = 0.7), t(678) = 4.018, *p* = 0.00, and individuals residing in quarantined areas (M = 4.6, SD = 0.6) appear to be more compliant than those from unrestricted areas (M = 4.5, SD = 0.6) t(1188) = 2.319, *p* = 0.02.

Furthermore, the level of compliance can be observed to be directly proportional to the age of the participants (Spearman’s rho = 0.134, *p* = 0.00), but also directly proportional to their level of education (Spearman’s rho = 0.141, *p* = 0.00).

In the context of the effects of the health crisis on the population’s behavior, two categories of behavioral change were identified (Kaiser-Meyer-Olkin Measure of Sampling Adequacy = 0.728, *p* < 0.00, Table 7): change in daily domestic behavior (home sanitation and the increase of personal hygiene measures, a better management of household financial and consumption behaviors, respectively a greater attention towards the consumption of health information) and changes in the practices of mobility and daily interaction (fewer meetings with relatives and friends, and fewer trips outside the household):Factor 1—daily domestic behavior explains 38.6% of the varianceFactor 2—spatial mobility and daily interactions account for 24.2% of the variance

As expected, the behavioral changes experienced by individuals residing in quarantined localities (M = 3.5, SD = 0.7) with regards to mobility and interaction are more prominent when compared to those of individuals residing in unrestricted areas (M = 3.3, SD = 0.7), t(1227) = 3.725, *p* = 0.00, a finding that could be explained both as a result of compliance with the restrictions, and as a component of the internal motivation of the participants.

The presence of a directly proportional relationship between the age of the respondents and the intensity of behavioral changes with regards to the frequency of meetings with friends and relatives can also be observed (Spearman’s rho = 0.06, *p* < 0.02). This positive correlation highlights that, the older the participants, the lower their interaction with relatives and friends is, decreasing to the point of avoiding this type of interaction for a certain period of time. Similarly, the increase in the age of participants is directly associated with the manifestation of interest in obtaining health-related information and news (Spearman’s rho = 0.12, *p* = 0.00), which indirectly highlights the concern of the population (e.g., as age increases) with regards to the development of the COVID-19 outbreak.

## 4. Discussion

The epidemiological crisis in Suceava started with a very large number of illnesses among the medical staff, a situation that accentuated the concern among the population. Impossibility of preventing the occurrence and the management of this outbreak through local resources are factors that have contributed significantly to establishing a negative public perception towards the poor management of SCH, and less towards the responsibility of medical staff, their inappropriate behavior, and the non-compliance with protection measures.

However, against this general background, the SCH enjoyed national recognition before the outbreak of the crisis, as a result of significant investments through which it had been modernized. The 50 million euros investments made the hospital rather attractive to health personnel, in the context in which 134 doctors were hired by the institution between 2009 and 2019 [47]. Moreover, the hospital was decorated with the “Sanitary Merit” Order and Knight Rank in April 2019, as a token of appreciation “for the results obtained in the improvement of the medical act”. The progress in increasing the quality of health care, paired with the existing discrepancy between the services offered by SCH on the one hand, and the other hospitals in the county on the other hand, pushed the health unit in Suceava towards becoming the main health center attraction of the county, an attraction which applied to both permanent residents, as well as those returning temporarily from EU countries, particularly during the summer months, overcrowding the Emergency Reception Unit. In the context of increasing rates of migrant return from Italy and Spain as a result of the pandemic, it can be concluded that this particularity has also contributed to the onset of the crisis in Suceava.

Human mobility has been acknowledged to have the potential to influence the spread and intensification of an epidemic [48], with existing research highlighting that the role that population movement can play in the community transmission of the virus is undeniable, and that different migration intensity can determine spatial variations in the manifestation of the epidemic [8,9,49]. The return of migrants settled in Italy and its influence on the spread of COVID-19 in Romania has been analyzed and demonstrated in several studies [50,51,52,53]. Romania encountered a specificity that other European countries did not, with tens of thousands of migrants not being accounted for in the official statistics. According to the Sociopol survey, which was conducted between the 18th and 23rd of March 2020, approximately 800,000 Romanians returned from abroad, with half of them admitting having returned to Romania out of fear of becoming ill. Moreover, the Ministry of Internal Affairs informed in March that roughly 10,000 people had returned to Suceava County, quickly becoming vectors of community transmission. The non-declaration of the locality and country of origin, and the lack of adherence to quarantine measures by those returning were issues frequently indicated by the authorities and the media, in an attempt to explain the outbreak in Suceava, consequently blaming the Romanian Diaspora returning to the country. The reiterated interventions of the ministers of health and interior, the president’s speeches, and the distribution of an exceptionally large volume of information through news channels, regardless of their veracity and quality, have considerably amplified the concerns of the population. The public attitudes and understanding of the causes that have generated the outbreak have, at least to a certain extent, been altered by the media. In fact, almost half of the respondents consider that the central press depiction of the situation in Suceava was exaggerated, even biased, and over a quarter of the participants share the same views about the reports in the local press.

The responsibility of medical staff in triggering the crisis is, in the perception of the population, a secondary factor. It is interesting that the level of trust of the population in the medical staff has similar values between the two moments: before the beginning of the pandemic, as well as after the cessation of this challenge.

This situation can be interpreted as a proof of the fact that the population does not consider the medical staff as the main responsible for triggering the crisis, but transfers the responsibility on the management and local decision-makers. In addition, the very large number of Romanians coming from abroad and the request for medical services from SCH along with their dishonest behavior in the sense of not declaring their return from the red zones were mitigating circumstances for the medical staff.

The transfer of responsibility from medical staff to SCH management probably also stems from the mismatch between the positive image of SCH (which generated a high level of expectations) and its inability to manage the crisis, which amplified the feeling of insecurity and lack of confidence.

Regarding the attitude of the population towards the measures imposed by the authorities, it is interesting that the dissatisfaction was mainly towards the local authorities, although the harshest measures were imposed by the central authorities. A possible explanation can be attributed to the fact that the quarantine decision was the result of the lack of reaction of the local authorities. The negative perception of the population is related to the involvement of the political factor in the management of the SCH and to the widespread corruption (“… appointments based on political criteria …”, “… employment and …”, “… involvement politics … in the management of the hospital and its transformation into an electoral platform in elections”). Against the background of a general behavior marked by the observance of the imposed restrictions, the results of the study showed that compliance with the imposed rules reflects the concern of the population with regards to their health and vulnerability rather than their trust in authorities.

Undoubtedly, the difficulty of managing this health crisis has highlighted the inefficiency of poor management, unable to mobilize the necessary human and financial resources and a politically subservient medical system [54]. Local peculiarities, related to mass remigration, have amplified the scale of the crisis.

## 5. Conclusions

The emergence and development of the epidemiological outbreak in Suceava is a specific situation that falls within the global pandemic context. Beyond the mismanagement of the crisis, each participant appears to have acknowledged that the current period requires fundamental changes in daily life activities and the adjustment of attitudes, both at the individual and collective level.

Subjected to the harshest measures to limit the spread of the outbreak, the population of Suceava County experienced the limitations of a poorly managed medical system, and correctly identified a series of shortcomings, valid for the Romanian medical system in general. Despite the perception of the population having changed as a result of fear and freedom of movement deprivation, it accurately indicated issues of underfunding, corruption, and the strong exodus of medical staff as the main triggers of the epidemiological crisis. Furthermore, the negative opinion towards the local decision-makers, the lack of efficient reaction, the involvement of the political factor in the management of SCH and the corruption present at the level of all structures are also issues highlighted by the perception of the sample used.

The crisis in Suceava was the result of the first confrontation of the Romanian health system with a new and difficult situation, which required awareness, responsibility and competent solutions. Political involvement in management, corruption and underfunding in the system were not present only in SCH, they have long been reported throughout the country. The novelty of the situation and the lack of anticipation of the effects of remigration on a single provider of high-performance health services at county level by poor management have made Suceava the “scapegoat” of a system with many dysfunctions acquired and amplified in the last 30 years. The fact that this crisis was a hard lesson both for Suceava and for the entire Romanian health system is proved by the absence of similar crises in the following months, even against the background of a much larger number of cases and similar conditions (return of Romanians from abroad during the summer holidays and winter holidays).

At the level of the population, however, the situation is more nuanced. The intensification of arrivals from abroad associated with the specific practices of the winter holidays seem to diminish the effects of the experience that the county went through in the spring of 2020. Even though the peak of the crisis in Suceava seemed to have been overcome (the county was, in December 2020, on the 14th place at national level according to the total number of infections and on the 30th place according to no. of infections per 1000 inhabitants), currently (January 2021), the number of new daily cases places the county among the top 5 nationally, and the city is one step closer to re-entering the red scenario (with the harshest restrictions).

The management of the crisis in Suceava required a fundamental change in the daily way of life, with the most unbearable restrictions being related to the spatial mobility of the population and to their socialization practices. The COVID-19 threat highlights that the burden of strict measures, if legal and guided by scientific expertise, can be accepted by the general public, if it is required by a profoundly grave situation or imposed by an authority that generates trust. The population is capable of large-scale empathy, the abandonment of certain individual freedoms and the adoption of a resilient way of life, if it serves the personal and general good.

Within this context, the crisis in Suceava can be considered the first local manifestation of a medical system crippled by various deficiencies, particularized by the high rate of those who returned from infected areas, in the context of the pandemic; it could represent the starting point of fundamental reforms in the Romanian medical system, which are yet to be observed. The first manifestation of the pandemic in Romania can be seen as an opportunity to learn valuable lessons, both for the health system and for society in general, similar to other severely affected European countries; the present study can therefore be the starting point in the analysis of the way in which these lessons are assimilated by the Romanian medical system and relevant policy makers.

## Figures and Tables

**Figure 1 ijerph-18-01406-f001:**
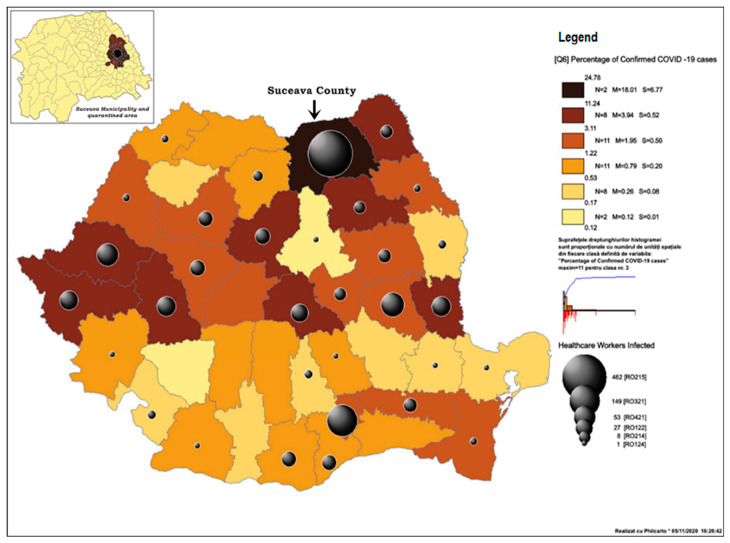
Confirmed COVID-19 cases per county (GCS, 16th of April 2020).

**Table 1 ijerph-18-01406-t001:** Factor Analysis Table for the public perception of causes that contributed to the appearance of the crisis in Suceava County; Rotated Component Matrix ^a^.

	Component			Communalities
	F1	F2	F3	
The political appointment of hospital leadership lacking necessary professional qualifications and competencies	0.892			0.824
Corruption at the level of local leadership structures	0.875			0.821
The continuation of informal pay (bribery), despite significant wage increases and the conditioning of the medical act	0.589		0.497	0.601
The lack of adherence to safety measures by citizens		0.891		0.797
The dishonest behavior of individuals who did not provide accurate information about being exposed to contamination risks		0.776		0.636
The lack of firm sanctioning (by local police and authorities etc.) of individuals who did not respect the restrictions imposed		0.642		0.485
The inadequate attitude of medical staff towards the growth risk of the epidemic			0.868	0.809
Deviations from professional conduct of medical staff			0.864	0.783
% of variance	25.14	23.58	23.20	

^a^ Extraction Method: Principal Component Analysis. Rotation Method: Varimax with Kaiser Normalization. Rotation converged in 4 iterations.

**Table 2 ijerph-18-01406-t002:** Factor Analysis Table for the public perception of the high incidence of infection amongst healthcare professionals; Rotated Component Matrix ^a^.

	Component		Communalities
	F1	F2	F3
The lack of awareness towards the gravity of the situation	0.829		0.712
The negligence in applying protective measures	0.813		0.688
The lack of professionalism exhibited by medical staff	0.758		0.592
Ignoring the danger posed by the large number of patients returning from red zones	0.753		0.655
The lack of protective equipment		0.804	0.647
The psychological pressure on the medical staff		0.680	0.466
The dishonesty exhibited by patients returning from red zones		0.671	0.542
% of variance	36.91	24.53	

^a^ Extraction Method: Principal Component Analysis. Rotation Method: Varimax with Kaiser Normalization. Rotation converged in 3 iterations.

**Table 3 ijerph-18-01406-t003:** Table of associations: quarantined locality and factors.

Variables	Quarantined Locality			χ^2^	df	*p*
	N	Yes	No			
Deviations from the professional conduct of the medical staff				9.41	1	0.002
To a small extent	421	200	221			
To a large extent	766	435	331			
The continuation of informal pay (bribery) and conditioning the medical act				4.294	1	0.038
To a small extent	342	200	142			
To a large extent	820	425	395			
Underfunding of the Romanian health system				5.493	1	0.019
To a small extent	295	176	119			
To a large extent	886	459	427			
Non-compliance of citizens with security measures				9.017	1	0.003
To a small extent	220	139	81			
To a large extent	971	505	466			

**Table 4 ijerph-18-01406-t004:** Description of measures implemented by authorities.

Difficulties	*N*	Mean	Std. Dev.
… restricted travel outside the quarantined area	1219	1.83	0.912
… limited meetings with friends and family	1230	1.86	0.882
… limited freedom to travel for recreational and social activities	1229	2.04	0.903
… limited access to public spaces (parks, markets, churches, etc.)	1229	2.14	0.964
…restrictive measures prohibiting groups bigger than 3–4 to meet and gather	1226	2.31	0.967
… limited freedom to travel for fitness activities	1221	2.33	0.948
… use of protective equipment in public spaces (gloves, masks, etc.)	1225	2.63	0.940
… limited access to “non-essential” shopping	1231	2.65	0.832

**Table 5 ijerph-18-01406-t005:** Attitude towards the measures adopted by authorities and quarantined locality (Independent test T).

Variables: Limitation of	*N*	Mean	Std. Dev.	*t*	df	*p*
… travel for recreational and social activities				3.270	1173.8	0.001
Quarantined area	661	1.96	0.885			
Non-quarantined area	568	2.14	0.914			
… travel for fitness activities				−3.057	1219	0.002
Quarantined area	654	2.25	0.948			
Non-quarantined area	567	2.42	0.940			
… travel outside the quarantined area				−5.808	1197.8	0.000
Quarantined area	660	1.69	0.917			
Non-quarantined area	559	1.99	0.881			
… travel for “non-essential” shopping				3.270	1173.8	0.001
Quarantined area	662	2.73	0.803			
Non-quarantined area	569	2.57	0.857			
… meetings for groups larger than 3–4 people				2.111	1206.5	0.035
Quarantined area	659	2.37	0.980			
Non-quarantined area	567	2.25	0.950			

**Table 6 ijerph-18-01406-t006:** Table of associations: restrictive measures by age of respondents.

Variables: Limitation of	*N*	Age Group (Years)			χ^2^	df	*p*
		<25	25–50	>50			
… travel for shopping					8.310	2	0.016
Difficult	483	111	291	81			
Not difficult	748	129	508	111			
… travel for fitness activities					10.652	2	0.005
Difficult	686	120	473	93			
Not difficult	535	118	321	96			
Wearing protective equipment in public spaces					6.441	2	0.040
Difficult	530	95	338	97			
Not difficult	695	144	459	92			

**Table 7 ijerph-18-01406-t007:** Factor Analysis Table for the public perception regarding of population’s behaviour changes; Rotated Component Matrix ^a^.

Behavioral Change	Component		Communalities
	F1	F2	
Cleanliness and home sanitation	0.817		0.669
Personal hygiene measures	0.755		0.573
Management of household financial resources	0.713		0.522
Food consumption behavior	0.703		0.527
Information/health related news consumption	0.653		0.449
Frequency of social interactions		0.903	0.829
Frequency of ordinary travel		0.897	0.826
% of variance	38.58	24.20	

Extraction Method: Principal Component Analysis. Rotation Method: Varimax with Kaiser Normalization. ^a^ Rotation converged in 3 iterations.

## Data Availability

The data presented in this study are available upon request from the appropriate author. The data is not available to the general public. The approved public, with competences in the analysis and interpretation of data from sociological studies will be able to request access to the study database.

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
