# Peer review of "Public Perception of the First Major SARS-Cov-2 Outbreak in the Suceava County, Romania"

_ijerph, 2021, doi:10.3390/ijerph18041406_

Round 1

Reviewer 1 Report

This is a generally well-written paper, and the aim is to understand the 'hotspot' of Covid-19 infections in Suceava County, Romania. The introduction is very well written, cites relevant studies, and clearly presents the research questions and aim(s) of the paper. However, after this first section, the paper becomes chaotic and hard to follow. I will not detail all the concerns but focus on the main points.

For one, the methodology is (much) too concise and important information is lacking. For example, there was a 33-item measure administered to a general population sample but there is no information about:

  • how the sample was recruited
  • inclusion/exclusion criteria
  • ethics approval
  • the content of the measure/questionnaire; not only the questions related to the Covid-19 crisis and the specific context of Suceava, but also with regard to which demographics were collected. In other words, which questions were asked? 
  • How the KAP (Knowledge, Attitude, Practice) model was translated/used to design the questionnaire
  • data analysis plan
  • etc etc

Because all this crucial information is missing, it is hard to understand the authors' approach to data analysis in their results section. This section seems to be a mix of descriptives and psychometric analyses (factor analyses), but it is hard to determine where the authors actually investigate the research questions they posed in their introduction. There is a large number of tables which in this case hinder rather than help interpretation of the text.

Furthermore, and perhaps most importantly, I do not think the authors interpret their analyses correctly. For example, on page 7 they state "highlighting that the degree of association with internal causes those related to SCH activity), considered the main culprits for the situation created, increased in
older participants (Table 3)" --> but Table 3 presents a t-test, which only signifies differences between (age) groups, not the increase or decrease of associations. All in all, I strongly doubt the authors' approach to data analysis and their interpretation of their findings.

I think this manuscript needs a thorough revision before it can be considered for publication, and that especially the methodology and results sections need to be revised. The content of the questionnaire needs to be provided, the psychometric (validation & reliability) analyses need to be conducted before the authors perform statistical tests - and the reason for conducting these tests needs to be argued and interpreted correctly. The results need to be presented much more concisely and the specific analysis approach should be guided by the test's ability to answer the research questions.

Author Response

Dear Reviewer,

We modified the manuscript according to all observations/remarks received. We accomplished all the requirements. Please find below the answer to your observations.

Thank you!

Despina Saghin and the co-authors

Reviewer 2 Report

I only suggest that the authors deepen the analysis of the results according to the gender of the participants.

Author Response

(The authors gave the same response as above.)

Reviewer 3 Report

The paper tries to explain, why the city and county of Suceava was characterised by the highest relative number of infections nationwide and also by the highest relative number of infected health professionals.

The main source of data for this analysis is the questionnaire. Seems that the sample may be representative, but authors do not confirm this in methodology part. Based on questionnaire, too high number of tables is created, making the text difficult to follow (some of tables are not connected to stated RQs). I recommend to authors to deal with most critical elements discovered.

The critical weakness of this paper is missing discussion. Opinions of citizen are important, but should be compared with facts, at least to some extent. Did the behaviour of medical staff in Suceava differ from other hospitals in the country? Is corruption at the level of local leadership structures different form other cities in the country? Was the level of non-compliance higher compared to other cities, etc. The authors should try to find clear arguments why Suceava suffered more than other cities – most existing studies do not suggest that behaviour of medical staff should be such factor. According to my opinion the most critical question stated by authors “Why did Suceava reach the peak number of Covid-19 cases in Romania?” (in spring! – is the situation now different?) – is not well responded.

Author Response

(The authors gave the same response as above.)

Round 2

Reviewer 1 Report

This manuscript has improved greatly, and the authors put a lot of work in revising, thank you for this. As a result, the aim and methods of the study is much clearer, and the analyses & results make more sense. There now is a discussion as well.

The only remaining comment I have is to condense the number of tables, e.g. by combining several of them into 1 larger table; adhere to reporting guidelines when doing so. Some information may be reported in supplemental materials, if it is not directly relevant for the content of the results & discussion. 

Author Response

 Dear Reviewer,

We modified the manuscript according to observation / remark received. We accomplished the requirement. Please find below the answer to your observations.

Thank you for all the comments/ remarks you have made, and we hope you agree with the publication of this article!

Despina Saghin and the co-authors

Reviewer 3 Report

OK

Author Response

                                              Dear Reviewer,

 Thank you for your consent to the publication of this article!

Despina Saghin and the co-authors
